# Preventive Effect of Spontaneous Physical Activity on the Gut-Adipose Tissue in a Mouse Model That Mimics Crohn’s Disease Susceptibility

**DOI:** 10.3390/cells8010033

**Published:** 2019-01-09

**Authors:** Florie Maillard, Emilie Vazeille, Pierre Sauvanet, Pascal Sirvent, Richard Bonnet, Lydie Combaret, Pierre Chausse, Caroline Chevarin, Yolanda Fernandez Otero, Geoffrey Delcros, Vivien Chavanelle, Nathalie Boisseau, Nicolas Barnich

**Affiliations:** 1Laboratoire des Adaptations Métaboliques à l’Exercice en conditions Physiologiques et Pathologiques (AME2P), Université Clermont Auvergne, F-63000 Clermont-Ferrand, France; Florie.MAILLARD@uca.fr (F.M.); pascal.sirvent@valbiotis.com (P.S.); yolanda.otero@valbiotis.com (Y.F.O.); Geoffrey.DELCROS@uca.fr (G.D.); vivien.chavanelle@valbiotis.com (V.C.); nathalie.boisseau@uca.fr (N.B.); 2Laboratoire Microbes Intestin Inflammation et Susceptibilité de l’Hôte (M2iSH), Université Clermont Auvergne, Inserm U1071, M2iSH, USC-INRA 2018, F-63000 Clermont-Ferrand, France; emilie.vazeille@uca.fr (E.V.); pierre.sauvanet@uca.fr (P.S.); rbonnet@chu-clermontferrand.fr (R.B.); caroline.chevarin@uca.fr (C.C.); 3Service d’Hépato-Gastro Entérologie, 3iHP, CHU, F-63000 Clermont-Ferrand, France; 4Service de chirurgie digestive, CHU, F-63000 Clermont-Ferrand, France; 5Department of Bacteriology, CHU, F-63000 Clermont-Ferrand, France; 6UNH, Unité de Nutrition Humaine, Université Clermont Auvergne, INRA, CRNH Auvergne, F-63000 Clermont-Ferrand, France; lydie.combaret@inra.fr; 7Laboratoire de Psychologie Sociale et Cognitive (LAPSCO), Université Clermont Auvergne, CNRS, F-63000 Clermont-Ferrand, France; pierre.chausse@uca.fr

**Keywords:** physical activity, mesenteric adipose tissue, Crohn’s disease

## Abstract

Crohn’s disease is characterized by abnormal ileal colonization by adherent-invasive *E. coli* (AIEC) and expansion of mesenteric adipose tissue. This study assessed the preventive effect of spontaneous physical activity (PA) on the gut-adipose tissue in a mouse model that mimics Crohn’s disease susceptibility. Thirty-five CEABAC10 male mice performed spontaneous PA (wheel group; n = 24) or not (controls; n = 11) for 12 weeks. At week 12, mice were orally challenged with the AIEC LF82 strain for 6 days. Body composition, glycaemic control, intestinal permeability, gut microbiota composition, and fecal short-chain fatty acids were assessed in both groups. Animals were fed a high fat/high sugar diet throughout the study. After exposure to AIEC, mesenteric adipose tissue weight was lower in the wheel group. Tight junction proteins expression increased with spontaneous PA, whereas systemic lipopolysaccharides were negatively correlated with the covered distance. *Bifidobacterium* and *Lactobacillus* decreased in controls, whereas *Oscillospira* and *Ruminococcus* increased in the wheel group. Fecal propionate and butyrate were also higher in the wheel group. In conclusion, spontaneous physical activity promotes healthy gut microbiota composition changes and increases short-chain fatty acids in CEABAC10 mice fed a Western diet and exposed to AIEC to mimic Crohn’s disease.

## 1. Introduction

Crohn’s disease (CD) is a chronic inflammatory bowel disease resulting from an aberrant immune response to intestinal microbiota stimulation in genetically predisposed individuals and/or under the influence of various environmental factors [1]. Patients with CD are characterized by a reduction of bacterial diversity [2] and dysbiosis, illustrated by the decrease of Firmicutes associated with the increase of Bacteroidetes and Proteobacteria [3]. Specifically, the microbiota of patients with CD shows a reduction in short chain fatty acid (SCFA)-producing species, such as *Faecalibacterium prausnitzii*, *Blautia faecis*, *Roseburia inulinivorans*, *Ruminococcus torques*, *Clostridium lavalense*, and *Bacteroides uniformis*, known for their anti-inflammatory and immune properties, and an increase of specific pathogenic *E. coli* strains, called adherent-invasive *E. coli* (AIEC) [4,5]. AIEC strains are strongly involved in CD etiology and bind to mannosylated carcinoembryonic antigen-related cell adhesion molecule 6 (CEACAM6), which is abnormally expressed on the surface of enterocytes in CD patients, leading to induction of intestinal inflammation [6]. Interestingly, in humanized transgenic CEABAC10 mice that express human CEACAM6, the AIEC reference strain LF82 colonizes and induces intestinal mucosa inflammation [7]. In these mice, the Western diet favors the host over-colonization by *E. coli* and AIEC strains [8]. Successful gut colonization by AIEC leads to the release of higher amounts of TNF-α, which aggravates inflammation [9,10]. Thus, transgenic CEABAC10 mice fed a Western diet and challenged with AIEC bacteria represent an original model to mimic CD susceptibility.

Although CD patients generally have a low or normal body mass index, the ratio of visceral adipose or abdominal subcutaneous adipose tissue to total adipose tissue is significantly higher compared with healthy controls [11]. In 1932, Dr. Burill Crohn described a significant presence of fat around the gut in these patients [12]. This expansion of mesenteric adipose tissue (visceral adipose tissue) can cover up to 50% of the diameter of the small intestine and colon in patients with CD [13]. These ectopic deposits are named “creeping fat” or “wrapping fat”. Visceral adipose tissue is an important producer of pro-inflammatory cytokines and chemokines (such as IL-6, TNF-α, and MCP-1) that contribute to systemic and intestinal inflammation [14]. Thus, adipose tissue is recognized as one of CD’s features [15], suggesting a cross-talk between adipose tissue, gut, and microbiota. Indeed, gut dysbiosis contributes to the impairment of intestinal permeability, favoring bacterial translocation [16]. This mechanism could explain the mesenteric adipose tissue expansion in CD [17,18].

Currently, the immunosuppressive or biological treatments used in CD are not curative and have many side effects. In this context, physical activity (PA) could be an attractive alternative and/or a complementary therapy due to its anti-inflammatory effects and capacity to decrease adipose tissue (including visceral fat mass). Several epidemiological studies have demonstrated a link between PA and reduced risk of developing CD [19]. Many studies in animal models of intestinal inflammation have highlighted the anti-inflammatory effect of voluntary exercise (i.e., using PA wheels). Specifically, PA prevents chemically-induced inflammation [20,21], intestinal injury following bacterial infection and/or exposure to bacterial components [22], and intestinal damage caused by high-fat and high-sugar diets (HF/HS) [23,24]. Gut microbiota can play a major role in these adaptations, and PA could be an original way to restore normobiosis in the context of chronic diseases. Indeed, changes in gut microbiota through PA may counteract the adverse effects of a high-fat diet [23] and induce protection following chemically-induced inflammation [20]. However, the effects of PA on the microbiota–adipose tissue cross-talk remain to be elucidated. Therefore, the aim of this study was to analyze the preventive effect of chronic exercise on this cross-talk in CEABAC10 mice fed a HF/HS diet. We hypothesized that spontaneous PA could (1) reduce total fat and mesenteric fat mass deposits and (2) promote beneficial changes in the intestinal microbiota.

## 2. Materials and Methods

### 2.1. Animals

FVB/N females and heterozygous CEABAC10 transgenic males (Charles River Laboratories) were mated in specific pathogen-free conditions in the animal care facility of the University Clermont Auvergne (Clermont-Ferrand, France). At 4–5 weeks of age, animals were weaned and genotyped. Eight-week-old CEABAC10 males (n = 35) were selected and randomly assigned to two groups: spontaneous PA on a wheel (wheel group; n = 24) and controls (n = 11). Animals were housed in individual cages with a reversed light-dark cycle in a temperature-controlled room (21 °C). All animal procedures were approved by the local ethics committee (APAFIS 3075-2015120813375547).

### 2.2. Spontaneous Physical Activity

Mice in the wheel group (n = 24) could perform spontaneous PA on the wheel in their individual cages. To measure the covered distance and speed, the wheels were equipped with a magnet and a sensor connected to a microcontroller digital input-output card (PIC18 4550 MICROCHIP). This card was programmed specifically for this use and allowed for the recording of each passage of the magnet in front of the sensor. This information was sent back to a computer for processing and storage using a specific program written in G language (LABVIEW National Instrument). Data analysis with the labChart version 7 software (ADInstruments, Sydney, Australia) allowed for calculation of the travelled distance (km) and the speed (m·min^−1^). Exercise was recorded continuously, and cages were visually checked at least four times/week.

### 2.3. Study Design

During the entire study, both groups were fed a HF/HS diet (13.2% proteins, 58.7% lipids, and 28.1% carbohydrates, primarily sucrose) (purified Diet 230 HF; Safe Diets, Augy, France) as a paradigm of the Western diet. The diet contained high levels of fat and simple sugars with an energy intake of 5317 kcal·kg^−1^ and a fiber content of 0.017%. As spontaneous PA can increase food intake [25], pair-feeding with controls was performed to ensure that the two groups consumed the same food quantity. After 12 weeks of a HF/HS diet, all mice were orally challenged with the AIEC LF82 strain (10^9^ bacteria), isolated from a patient with CD, for six consecutive days. Ten days after the first AIEC bolus, mice were sacrificed (Figure 1).

### 2.4. Weight and Body Composition

Weight was recorded weekly for 12 weeks (W0 to W12) and every day during bacterial exposure. Body composition was measured using the EchoMRI 3-in-1 instrument (Echo Medical Systems, Houston, TX, USA) at W0, W6, and W12 before bacterial exposure. Mesenteric fat pads were weighed *post-mortem*.

### 2.5. Oral Glucose Tolerance Test (OGTT) and Plasma Measurements

OGTTs were performed after 6h fasting at W0 and at W12 before bacterial exposure. Tail blood samples were taken at 0, 15, 30, 60, 90, and 120 min after oral gavage of glucose (1.1 g·kg^−1^ lean mass). Glycaemia was determined using a glucometer (Accu-chek Performa, Roche Diagnostics, Basel, Switzerland). The area under the curve for glucose (AUC) and the _net_AUC (obtained by subtracting the baseline glucose concentration) were calculated. Blood samples were centrifuged at 2000× *g* for 10 min and plasma samples were stored at −80 °C. Plasma insulin was measured using the Ultrasensitive Insulin ELISA Kit (ALPCO, Salem, NH, USA). The homeostasis model assessment of insulin resistance (HOMA-IR) index was calculated as follows: fasting insulin (mU·L^−1^) × fasting glucose (mmol·L^−1^)/22.5 [26].

### 2.6. Protein Extraction and Western Blotting

Colon tissue samples were homogenized in 500 µL of lysis buffer (25 mM Tris, 1 mM EDTA, 5 mM EGTA, 0.1 mM MgCl2, 10% glycerol, 150 mM NaCl, 1% Nonidet P-40, 1% SDS) supplemented with a protease inhibitor cocktail (cOmplete™, Mini, EDTA-free Protease Inhibitor Cocktail, Roche, Basel, Switzerland), 1 mM sodium orthovanadate, 1 mM PMSF, and 5 mM N-ethylmaleimide. Homogenates were centrifuged at 10,000× *g* at 4 °C for 5 min. Protein extracts were stored at −80 °C. The protein content was determined with a colorimetric assay (DC protein assay, Bio-Rad, Hercules, CA, USA). Then, 25 µg of proteins were separated on 12% SDS-PAGE gels, transferred to nitrocellulose membranes, and blocked with 5% bovine serum albumin in Tris-buffered saline (pH 8) containing 0.05% Tween 20 (TBS-T) at room temperature under agitation for 1h. Then, membranes were incubated with primary antibodies against occludin (1:500; 33-1500; Invitrogen, Camarillo, CA, USA) or zonula occludens-1 (ZO-1) (1:500; 61-7300; Invitrogen) at 4 °C under agitation overnight. After washes with TBS-T, membranes were incubated with secondary antibodies in TBS-T at room temperature under agitation for 1 h. Antibody interactions were detected with the Enhanced Chemiluminescence Detection Kit (RPN2108, Amersham Biosciences, Piscataway, NJ, USA). Images were acquired with the Bio-Rad ChemiDoc system and analyzed with Image J (1.50i version, National Institutes of Health, Bethesda, MD, USA). ZO-1, and occludin expression was normalized to GAPDH.

### 2.7. Mucosa-Associated Microbiota Composition Analysis by Illumina Sequencing

Colon samples were transferred in ZR BashingBead™ Lysis Tubes (0.1 and 0.5 mm, Zymo Research, Irvine, CA, USA) with lysis buffer (Maxwell^®^ RSC Buffy Coat DNA, Madison, WI, USA) and homogenized using a Precellys homogenizer (2 × 15 s followed by 2 min rest). Lysis tubes were centrifuged at 14,000 g (4 °C, 3 min), and supernatants were collected and centrifuged again to ensure that all beads were removed. Then, supernatants were loaded in the cartridges of a Maxwell^®^ RSC Instrument (Promega, Madison, WI, USA) for DNA extraction. DNA concentration was determined with a Qubit Fluorometer (Invitrogen), and DNA quality was evaluated by spectrophotometry (260/280 and 260/230 ratios, Nanodrop). The variable regions from V3 to V4 of bacterial 16S rRNA genes were amplified from purified DNA with MTP Taq DNA Polymerase, 10X MTP Taq Buffer (D7442-1500U, Sigma, Saint-Louis, MO, USA), and the following primers: forward CTTTCCCTACACGACGCTCTTCCGATCTACGGRAGGCAGCAG and reverse GGAGTTCAGACGTGTGCTCTTCCGATCTTACCAGGGTATCTAATCCT. All PCR amplifications were performed with the following cycling conditions: 94 °C for 1 min, followed by 30 cycles (94 °C for 1 min, 65 °C for 1 min, and 72 °C for 1 min), and a final elongation step at 72 °C for 10 min. Illumina sequencing was performed in collaboration with the GeT core facility (Toulouse, France). Paired-end read assembly, quality and length filtering, Operational Taxonomic Unit (OTU) picking (97% sequence identity threshold), and chimera removal were performed with UPARSE [27]. OTUs with low counts (<0.1% of the total number of sequences per sample) were excluded. Sequences of samples with over 6000 reads were loaded into the QIIME 1.9.1 pipeline for diversity analysis [28]. Taxonomy assignment was performed with the SILVA database 132 (https://www.arb-silva.de/). Alpha diversity of bacterial communities was assessed from four different indexes including richness and/or evenness (Chao1, Shannon, Simpson, and evenness). The Kruskal–Wallis test was used to estimate alpha diversity differences among groups. Beta diversity was used to analyze the dissimilarity among the groups’ membership and structure. Accordingly, abundance-weighted and/or phylogenetic-weighted distance matrices were generated on the basis of Bray–Curtis, and weighted/unweighted UniFrac distances were reported according to principal coordinate analysis (PCoA). Permutational analysis of variance (PERMANOVA with 999 permutations) was utilized to determine significant differences among groups. Significance testing for taxon abundance was accomplished with a Wilcoxon rank sum test and the Bonferroni procedure to correct *p*-values. The *p*-values < 0.05 were considered significant. A linear discriminant analysis effect size (LEfSe) analysis was performed to identify differentially abundant bacterial taxa among groups.

### 2.8. Fecal Short-Chain Fatty Acid (SCFA) Quantification

Weighted fecal samples were reconstituted in 100 µL Milli-Q^®^ water (Millipore, Billerica, MA, USA), disrupted, and then incubated at 4 °C for 2 h and centrifuged at 12,000× *g* at 4 °C for 15 min. Supernatants were weighed, and saturated phosphotungstic acid was added (1 g for 100 µL). After overnight incubation at 4 °C, samples were centrifuged again and SCFA concentrations were determined using gas chromatography (Nelson 1020, Perkin-Elmer, St Quentin en Yvelines, France), as previously described [29,30].

### 2.9. Plasma Lipopolysaccharide (LPS) Load Quantification

LPS was quantified using HEK-Blue-mTLR4 cells (Invivogen, San Diego, CA, USA), and plasma samples collected at sacrifice. Then, 180 µL of a cell suspension (1.4 × 10^4^ cells per mL in HEK-Blue Detection medium) (Invivogen) was added to 20 µL of each diluted (1:10) plasma sample. LPS (Sigma, St.Louis, MO, USA) was used as a positive control and standard range. Plates were incubated at 37 °C in 5% CO_2_ for 24 h, and alkaline phosphatase activity was measured at 620 nm.

### 2.10. Statistical Analysis

All statistical analyses were carried out using the Statistica software (version 12). Data are presented as the mean ± standard error of the mean (SEM). The data normal distribution was tested using the Kolmogorov–Smirnov test, and the homogeneity of variance was tested with the F-test. A two-way ANOVA with repeated measures was used to determine group (G) and time (T) effects and G × T interactions. When a significant effect was found, post-hoc multiple comparisons were done using the Newman–Keuls test. The non-parametric Mann and Whitney test was used to compare results between groups. Pearson correlations were used to test relationships between variables. The *p* values ≤ 0.05 were considered statistically significant.

## 3. Results

### 3.1. Spontaneous Physical Activity Decreases Total Fat Mass and Improves Glucose Metabolism

We first investigated the effects of 12 weeks of a HF/HS diet on body composition and glucose metabolism in exercised mice (wheel group; n = 21) and sedentary mice (control group; n = 10). Starting from W7 and up to W12, body weight was significantly lower in the wheel group than in controls, with the exception of W9 (*p* = 0.007) (Figure 2A). As total lean mass (expressed in %) was similar in the groups throughout the study period (Figure 2B), the decrease of total body mass in the wheel group was mainly due to a reduction in total fat mass compared with controls (Figure 2C; *p* ≤ 0.05).

Mice in the wheel group ran on average 3.58 ± 0.39 km per day at 8.53 ± 0.67 m·min^−1^ during the 12 weeks before bacterial exposition. However, the distance covered and the speed greatly varied among mice. Analysis of the distance and speed at W12 showed a significant correlation between these two parameters (Figure 2D; *r* = 0.4; *p* ≤ 0.05). Based on this observation, we systematically performed correlation analyses between metabolism/inflammation parameters and distance/speed to better understand the PA effects. Specifically, at W12, the average distance covered per day was negatively correlated with total fat mass (Figure 2E; *r* = −0.5; *p* ≤ 0.05) and positively correlated with total lean mass (data not shown; *r* = 0.5; *p* ≤ 0.01). Similarly, speed was positively correlated with total lean mass (data not shown; *r* = 0.5; *p* ≤ 0.05), but not to total fat mass.

Twelve weeks of spontaneous exercise and/or a HF/HS diet did not significantly change fasting plasma glucose, plasma insulin, and homeostatic model assessment of insulin resistance (HOMA-IR) index (Table 1). Similarly, plasma glucose values during the OGTT and _net_AUC values at W12 were not significantly different between groups (repeated measures ANOVA) (Figure 3A,B). However, fasting plasma glucose and _net_AUC were negatively correlated with the average distance covered per day (Figure 3C,D; *r* = −0.6; *p* ≤ 0.01 for both). The glucose _net_AUC was also negatively correlated with the average speed (Figure 3E; *r* = −0.6; *p* ≤ 0.01).

### 3.2. Spontaneous Physical Activity Prevents Mesenteric Adipose Tissue Accumulation

At W12, mice in both groups were orally challenged with AIEC LF82 bacteria for 6 days and sacrificed at day 10 after the first exposure (Figure 1). Body weight loss following AIEC exposure (percentage) during these 10 days was not significantly different between groups (Figure 4A), although it was lower in mice that ran faster (Figure 4B; *r* = 0.5; *p* ≤ 0.05). Moreover, the weight of mesenteric adipose tissue (reflecting visceral adipose tissue) was lower in the wheel group compared with controls (Figure 4C, *p* ≤ 0.05), particularly in mice that ran a greater distance (Figure 4D; *r* = −0.6; *p* ≤ 0.01).

### 3.3. Spontaneous Physical Activity Increases Tight Junction Protein Expression and Restores SCFA Production

A HF/HS diet alters intestinal permeability, favoring bacterial translocation and inflammation [10] and reducing SCFA production (acetate, propionate, and butyrate). At sacrifice, the expression of the colon tight junction proteins (occludin and ZO-1) was increased in the wheel group compared with controls (Figure 5A; *p* ≤ 0.05). Similarly, the concentration of propionate and butyrate in fecal samples was increased in the wheel group compared with controls (*p* ≤ 0.05), without effect for acetate (Figure 5B). Occludin, ZO-1, and butyrate levels were negatively correlated with mesenteric adipose tissue weight (data not shown, *r* = −0.5, *p* ≤ 0.01; *r* = −0.4, *p* ≤ 0.05; *r* = −0.4, *p* ≤ 0.05; respectively). Moreover, ZO-1 and occludin levels were positively correlated with butyrate levels (*r* = −0.5, *p* ≤ 0.01; *r* = −0.4; *p* ≤ 0.05; respectively). Finally, the plasma concentration of active LPS was comparable between groups (Figure 5C). However, the plasma concentration of active LPS was negatively correlated with the average distance (Figure 5E, *r* = −0.4; *p* ≤ 0.05).

### 3.4. Spontaneous Physical Activity Modulates the Mucosa-Associated Intestinal Microbiota

To investigate whether the changes in total and mesenteric fat mass, SCFA production, and tight junction expression in the wheel group could be related to gut microbiota modulation, we compared the abundance and composition of the colon mucosa-associated microbiota in the wheel group (n = 14) and control group (n = 7) at sacrifice. The α-diversity (Shannon index) did not show any difference in species richness between groups (Figure 6A). However, PCoA analysis based on the unweighted uniFrac distance matrices, which allows highlighting of the phylogenetic relationship and composition of colon bacterial microbiota samples, revealed that colon samples of the two groups clustered in two distinctive groups (Figure 6B). This indicated that spontaneous exercise changed the mucosa-associated microbiota composition, with principal component scores that accounted for 6.31% (PC1), 11.71% (PC2), and 30.97% (PC3) of the total variance. The significant separation of the wheel and control colon samples was confirmed with the ANOSIM test (*p* ≤ 0.05). As the β-diversity changes indicated a functionally modification of mucosa-associated microbiota, we performed a LEfSe analysis using the non-parametric Kruskal–Wallis rank sum test followed by linear discriminant analysis to identify genera that differed significantly between groups (Figure 6C). In line with the higher butyrate levels in the wheel group, *Ruminococcus* and *Oscillospira* (genus containing butyrate-producing colon species) were more abundant in the wheel group than in controls. Conversely, *Bifidobacterium* and *Lactobacillus*, which are recognized as health-promoting genera, were less abundant in the control group.

## 4. Discussion

In the present study, we decided not to use either Dextran Sodium Sulfate (DSS) or antibiotics treatment that allow AIEC to efficiently colonize and induce gut severe intestinal inflammation [7]. In this model, exposure to AIEC combined with a Western style diet was used to reflect what is happening in patients in remission to study primary prevention by physical activity and its impact on the mucosal-associated microbiota. Indeed, using a similar experimental design, we previously reported that a Western diet induces changes in gut microbiota composition, alters host homeostasis, and promotes AIEC gut colonization in genetically susceptible mice without inducing acute inflammation and histological damage [10]. Our results showed that spontaneous PA decreased total fat mass, and even if percentage of body weight loss was not significantly different between the two groups, mice that ran faster lost body weight more efficiently. PA also favored glucose metabolism in CEABAC10 mice fed a HF/HS diet. Moreover, in the wheel group, tight junction protein expression (reflecting intestinal permeability) was increased after AIEC bacterial exposure and this could contribute to reduced metabolic endotoxemia and could limit mesenteric adipose tissue expansion. In the present study, the increased expression of Occuldin and ZO-1 was not correlated with a significant translocation of bacterial LPS (correlation between LPS translocation and occluding or ZO-1 expressions, both: *r* = 0.04; *p* = 0.8). Thus, the increase in tight junction protein expression may precede strengthening of the intestinal barrier function. Spontaneous PA also induced changes in gut microbiota composition by increasing the proportion of SCFA-producing species belonging to *Oscillospira* and *Ruminococcus* genus, whereas beneficial and anti-inflammatory genera (*Bifidobacterium* and *Lactobacillus*) were decreased in controls. These changes were linked to higher fecal levels of propionate and butyrate in the wheel group.

For the last 10–15 years, *Escherichia coli* has been the most suspected bacteria in CD etiology. AIEC prevalence varies from 21% to 63% in patients with CD [5,31]. Moreover, a Western diet, characterized by low fiber and high fat intake leads to dysbiosis [10]. After exposure to the AIEC strain LF82, the group of CEABAC10 mice that ran the fastest lost the least weight. In our study, spontaneous PA did not alter the inflammatory state (data not shown), but increased tight junction protein expression, and the distance covered was negatively correlated with active LPS plasma levels. Moreover, mesenteric adipose tissue weight was lower in the wheel group compared with controls. Mesenteric adipose tissue expansion is a characteristic of inflammation in CD [15], and it is mainly due to bacterial translocation. Indeed, dysbiosis alters membrane junctions, thus increasing intestinal permeability [32]. The resulting bacterial translocation leads to metabolic endotoxemia (increase of LPS level) that is directly involved in the development of chronic inflammation and metabolic disorders, such as obesity, insulin resistance, and/or type 2 diabetes [33]. In our study, spontaneous PA increased tight junction protein expression, and the distance covered was negatively correlated with active LPS plasma levels. Moreover, tight junction protein expression was negatively associated with the amount of mesenteric adipose tissue, strengthening the importance of intestinal permeability in mesenteric adipose tissue expansion. This result is of great interest because it is thought that PA, especially high intensity and long duration exercise, can induce ischemia that increases intestinal permeability (only when at least 50% of blood flow is reduced). This phenomenon is called the “leaky gut”. Intestinal permeability is 1.5 to 3 times higher in high-level athletes than in recreational exercisers [34]. When duration and PA level are correctly controlled over a long period, exercise may be beneficial for gut health. Holland et al. found that in rats, exercise for 10 days (60 min at 30 m·min^−1^, 5 days/week) reduces intestinal inflammation 24 h post-exercise and suggested that exercise may contribute to the maintenance of normal intestinal permeability [35].

SCFA (propionate, butyrate, and acetate) are the main end products of intestinal microbial fermentation. They participate in intestinal barrier integrity by promoting the secretion of anti-inflammatory cytokines and anti-microbial peptides and by increasing the expression of tight junction proteins [36,37]. The present study found that spontaneous PA increased butyrate and propionate levels in feces. This could explain the higher tight junction proteins expression in the wheel group. This hypothesis is reinforced by the positive association between butyrate and tight junction proteins expression observed in the present study. In accordance with our findings, five weeks of spontaneous exercise increased n-butyrate levels in Wistar rats, but had no effect on acetate and propionate [38]. More recently, a study in C57BL/6 mice showed no effect of spontaneous exercise for 6 weeks on SCFA, but the butyrate/acetate ratio increased with exercise [20]. In humans, six weeks (3 days/week) of moderate (30 to 60 min at 60% of the heart rate reserve; HRR) to vigorous (60 min at 75% HRR) exercise increased SCFA in lean subjects, but not in volunteers with obesity [39]. In our study, butyrate level was negatively associated with mesenteric adipose tissue weight. Interestingly, G-protein coupled receptors (SCFA receptors) are found in adipose tissue [40], suggesting a role for SCFA in adipose tissue metabolism. Several studies reported that SCFA, especially acetate, decrease basal lipolysis in adipose tissue in humans [41] and also in cell lines [40,42]. Partial inhibition of basal lipolysis prevents ectopic fat accumulation and insulin resistance and concomitantly increases SCFA-induced oxidation. In this context, the increase of butyrate and propionate levels may partly explain the lower weight and total fat mass and the activity intensity-related improvement of glucose metabolism observed in our study.

In recent years, the effects of PA on gut microbiota have been increasingly studied in animals and also in humans. Here, β-diversity was significantly different between groups, partly due to a decrease of *Lactobacillus*, *Bifidobacterium*, *Prevotella*, *Turicibacter*, *SMB53*, and *Brachyspira* in the control group and an increase of *Anaerotruncus*, *parabacteroides*, *Desulfovibrionaceae*, *Oscillospira*, and *Ruminococcus* in the wheel group. *Lactobacillus* and *Bifidobacterium* are two health-promoting bacteria used as probiotics to counteract the effect of a Western diet [43,44]. Moreover, they are reduced in patients with CD [45]. The decrease of *Lactobacillus* and *Bifidobacterium* in the control group suggests that PA prevents the HF/HS diet-effects partly by modulating the gut microbiota. Similarly, Evans et al. suggested that 12 weeks of spontaneous PA in a mouse model of obesity could play a role in the prevention of adverse Western diet-effects [23]. *Oscillospira* and *Ruminococcus* are negatively associated with visceral fat mass and the android/gynoid ratio, suggesting a potential proactive role in cardiovascular risk [46]. Moreover, the abundance of *Ruminococcus*, a butyrate producer, is reduced in patients with CD [47]. Our study indicates that spontaneous PA can increase their abundance in the colon of CEABAC10 mice that mimic CD susceptibility. *Oscillospira* is an anaerobic bacterial genus from the Clostridial cluster IV and are part of the Firmicutes phylum. *Oscillospira* abundance has been associated with leanness and is reduced in patients with CD [48,49]. The benefits of *Oscillospira* are partially due to butyrate production [50]. Petriz et al. showed a positive association between *Oscillospira* and blood lactate levels, suggesting an exercise-intensity effect on gut microbiota composition [51]. The increase of *Oscillospira* following lightly forced exercise (8–12 m∙min^−1^; 5% grade) compared with voluntary exercise supports this hypothesis [52]. However, in the first study, small numbers of animals (n = 9) were used, and in the second work, the covered distance and speed in the spontaneous activity wheel group were not recorded. Therefore, the effect of PA, especially its duration and intensity, on *Oscillospira* abundance needs additional investigation. In our study, *Oscillospira* and *Ruminococcus* could have contributed to an increase in the barrier-function integrity by reducing LPS translocation and adipose tissue expansion through butyrate production.

In conclusion, this study shows that spontaneous PA promotes colon mucosa-associated microbiota composition changes in CEABAC10 mice fed a HF/HS diet and challenged with bacteria. The abundance of beneficial and anti-inflammatory genera (*Bifidobacterium* and *Lactobacillus*) was reduced in the control group, whereas the abundance of butyrate producers that are considered health-related genera (*Oscillospira* and *Ruminococcus*) was increased in the spontaneous PA group. These findings suggest that gut microbiota changes could be an early event before inflammation. In addition, the modulation of gut microbiota by regular PA appears as an attractive complementary therapeutic strategy to extend the periods of remissions in patients with CD.

## Figures and Tables

**Figure 1 cells-08-00033-f001:**
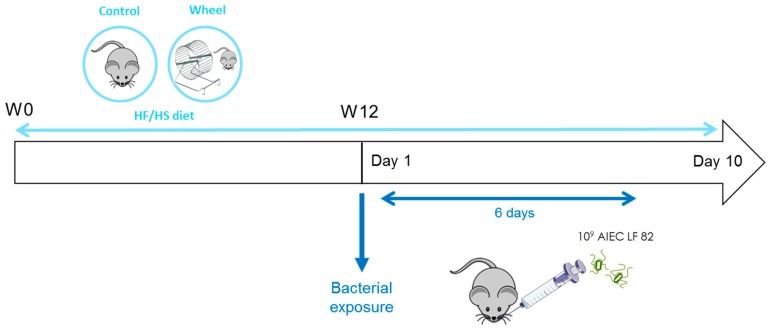
Study protocol. Eight-week-old mice were subdivided in two groups: animals that did spontaneous physical exercise on a wheel (n = 24) and controls (n = 11). The covered distance and speed in the wheel group were recorded continuously. At week 12 (W12), both groups were exposed to adherent-invasive *E. coli* (AIEC) LF82 for 6 days and were sacrificed 4 days later. Animals were fed a high fat/high sugar diet (HF/HS) throughout the study, and pair-feeding was performed during the first twelve weeks.

**Figure 2 cells-08-00033-f002:**
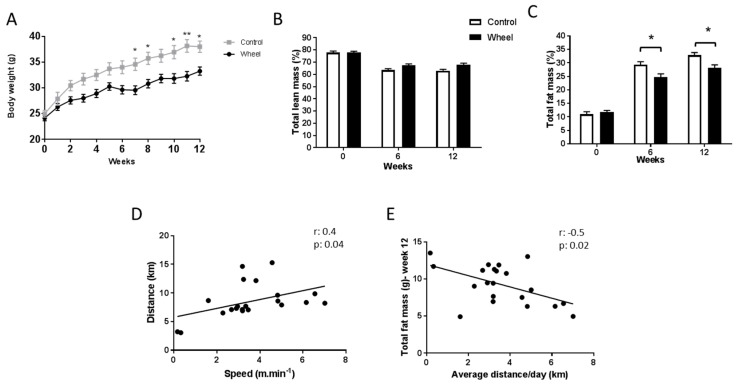
Effect of 12 weeks of spontaneous physical activity on body weight (**A**), total lean mass (%) (**B**), and total fat mass (%) (**C**). Correlations between covered distance and speed (**D**) and total fat mass and average distance/day (**E**). Data are expressed as the mean ± SEM. * *p* ≤ 0.05; ** *p* ≤ 0.01.

**Figure 3 cells-08-00033-f003:**
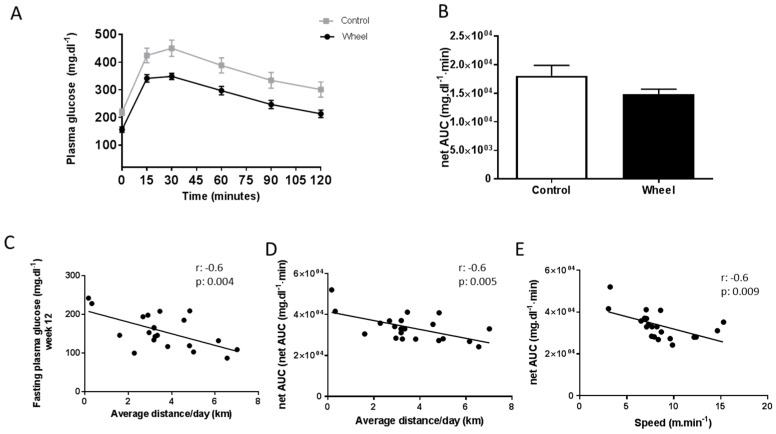
Effect of 12 weeks of spontaneous physical activity on blood glucose response to an Oral Glucose Tolerance Test (OGTT) (**A**), and glucose _net_AUC (**B**). Correlations between fasting plasma glucose and average distance/day (**C**), glucose _net_AUC and average distance/day (**D**), and glucose _net_AUC and speed (**E**). Data are expressed as the mean ± SEM. AUC = area under curve.

**Figure 4 cells-08-00033-f004:**
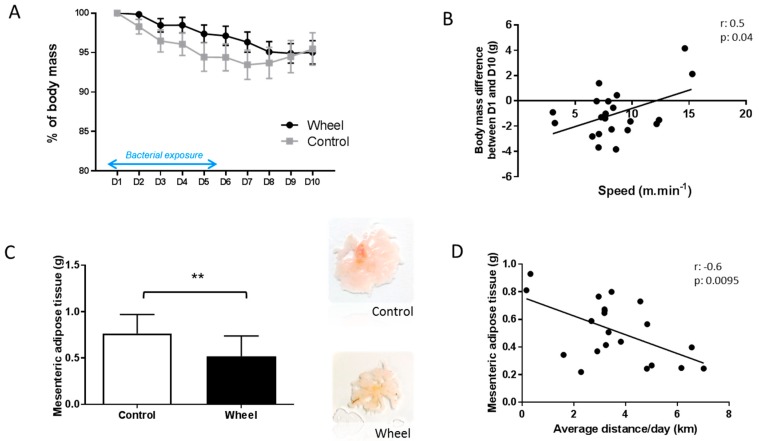
Effect of 12 weeks of spontaneous physical activity on body mass and body composition after bacterial exposure. Body mass loss (%) during 10 days (**A**). Correlation between body mass difference between day 1 (D1) and D10 and speed (**B**). Weight of mesenteric adipose tissue measured post-mortem (**C**). Correlation between mesenteric adipose tissue weight and average distance/day (**D**). Data are the mean ± SEM; ** *p* ≤ 0.01.

**Figure 5 cells-08-00033-f005:**
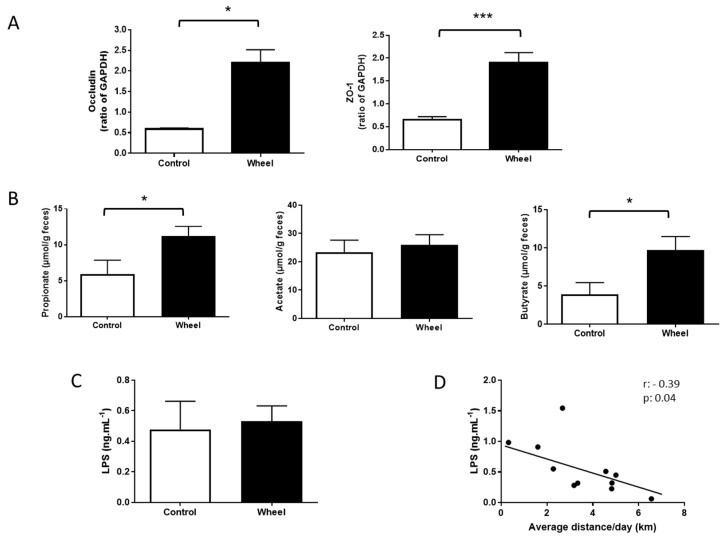
Effect of spontaneous physical activity on colon tight junction protein expression (**A**) on the fecal concentration of the three main short-chain fatty acids (**B**) and on the plasma concentration of active lipopolysaccharide (LPS) (**C**). Correlation between active LPS concentration and average distance/day (**D**). Data are the mean ± SEM; * *p* ≤ 0.05; *** *p* ≤ 0.001.

**Figure 6 cells-08-00033-f006:**
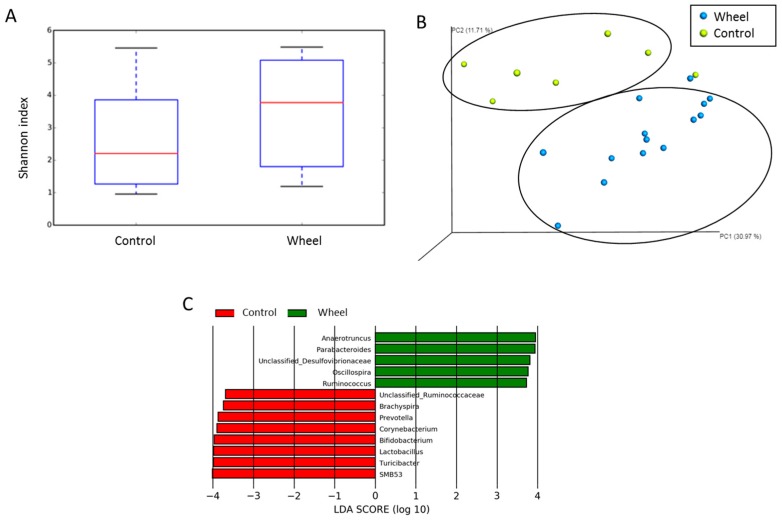
Mucosa-associated microbiota composition analyzed by16S rRNA gene sequencing using colon DNA samples (Illumina MiSeq system) at the end of the study (n = 14 animals from the wheel group and n = 7 controls). Shannon index (**A**), PCoA plots (**B**), and LEfSe analysis (**C**).

**Table 1 cells-08-00033-t001:** Glucose profile change (difference = 12 weeks–baseline).

	Controls	Wheel Group
**Fasting blood glucose (mg∙dL^−1^)**	60.0 ± 13.5	30.5 ± 12.8
**Serum insulin (ng∙mL^−1^)**	0.9 ± 0.6	1.2 ± 0.2
**HOMA-IR**	19.4 ± 10.1	13.8 ± 2.5

HOMA-IR: Homeostatic model assessment of insulin resistance. Data are expressed as the mean ± SEM.

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
