# Peer review of "Preventive Effect of Spontaneous Physical Activity on the Gut-Adipose Tissue in a Mouse Model That Mimics Crohn’s Disease Susceptibility"

_cells, 2019, doi:10.3390/cells8010033_

Round 1

Reviewer 1 Report

Manuscript Cells-409008 “Preventive effect of spontaneous physical activity on the gut-adipose tissue in a mouse model that mimics Crohn’s disease susceptibility”. This paper reports the effect of spontaneous physical activity in intestinal barrier and mesenteric adipose tissue in a mouse model of Crohn’s disease.

The manuscript is well focused with very good experimental design. The authors have employed a series of sound analytical techniques and protocols and provide a significant amount of reliable data, which support the discussion and are presented in a reader-friendly way. Overall it is an interesting and well-organized manuscript.

Minor comments

Data about diet consumption are not presented, this is important since it could create differences between groups. The diet composition is not well specified, it should be specified that the diet composition is respect to the total of energy provided.

The sample size changes during the study without any reason. At the beginning of the protocol description, the wheel group consists of 24 mice and the control of 11, in the exercise protocol there are only 21 mice in the exercise group and 10 in the control group. In the sequencing data there are only 14 exercise mice and 7 controls. This should be explained.

Many of the results are based on correlation analyses, however, these results do not really show differences between the experimental groups. A stratified analysis based on the amount of physical exercise performed would be more appropriate to show results between groups.

Results show an increase in tight junction and no changes in the amounts of LPS in the blood. Why do these differences occur?. A paragraph on this topic should be included in the discussion.

The Subtitle 3.3 “Spontaneous physical activity improves the barrier function and restores SCFA production” creates confusion because the barrier function as such has not been really measured.

Crohn's disease mainly affects the ileum. Why microbiota and intestinal permeability determinations have been performed on the colon?

Author Response

Point by point replies to the Editor and to the Reviewers’ comments

We are grateful to the Chief Editor and the Reviewers for their thoughtful and constructive comments. Below, we have replied to the issues raised in a point-by point manner, and we revised our manuscript accordingly. Thank you for considering this revised manuscript.

Comments and Suggestions for Authors: reviewer 1

Manuscript Cells-409008 “Preventive effect of spontaneous physical activity on the gut-adipose tissue in a mouse model that mimics Crohn’s disease susceptibility”. This paper reports the effect of spontaneous physical activity in intestinal barrier and mesenteric adipose tissue in a mouse model of Crohn’s disease.

The manuscript is well focused with very good experimental design. The authors have employed a series of sound analytical techniques and protocols and provide a significant amount of reliable data, which support the discussion and are presented in a reader-friendly way. Overall it is an interesting and well-organized manuscript.

Minor comments

Data about diet consumption are not presented, this is important since it could create differences between groups. The diet composition is not well specified, it should be specified that the diet composition is respect to the total of energy provided.

A « pair feeding » was performed on the control group, controlled every day (to check that the animals did not run out of food) and the diet was changed every 2 days. No difference between the two groups was observed concerning energy intake (see additional figure below and lines 119-120 in the revised version).

Concerning the diet composition, in line 114 in the revised version: During the entire study, both groups were fed a HF/HS diet (13.2% of proteins, 58.7% of lipids and 28.1% of carbohydrates, primarily sucrose) (purified Diet 230 HF; Safe Diets, France), as a paradigm of Western diet (lines 115-117).

The sample size changes during the study without any reason. At the beginning of the protocol description, the wheel group consists of 24 mice and the control of 11, in the exercise protocol there are only 21 mice in the exercise group and 10 in the control group. In the sequencing data there are only 14 exercise mice and 7 controls. This should be explained.

The reviewer is right to raise this point. At the beginning of the protocol, the wheel group consists indeed of 24 mice and the control of 11, but some mice died (n=3) in the wheel group, and n=1 in the control group, explaining the numbers of 21 and 10 at the end of the protocol. Regarding the analysis of the musoca-associated microbiota, we kept only the samples for which the sequencing results passed all the quality controls. For some samples, we did not have enough reads to correctly analyze the composition of global microbiota and these samples were scratched so as not to bias the results.

Many of the results are based on correlation analyses, however, these results do not really show differences between the experimental groups. A stratified analysis based on the amount of physical exercise performed would be more appropriate to show results between groups.

Eight-week-old mice were subdivided in two groups at the beginning of the protocol: animals that did spontaneous physical exercise on a wheel (n=24) and controls (n=11). The covered distance and speed in the wheel group were recorded continuously but the spontaneous activity of the controls was not. Therefore, an ANCOVA between the two groups was not possible and the use of correlations between the distance performed or the mean speed and the metabolic or anthropometric changes appeared to us as the best solution to demonstrate that the level of physical activity may alter health related parameters. 

Results show an increase in tight junction and no changes in the amounts of LPS in the blood. Why do these differences occur? A paragraph on this topic should be included in the discussion.

The increase in tight junction protein expression may precede the increase in intestinal permeability. As such, an increase in the expression of certain tight junction proteins could represent a first step in strengthening intestinal barrier function, but was not necessarily associated with an immediate decreased of bacterial LPS translocation (correlation between LPS translocation and occludin expression: r=0.04; p=0.8 and correlation between LPS translocation and ZO-1 expression: r=0.04; p=0.8). This point has been added in the discussion section (lines 338 to 342).

The Subtitle 3.3 “Spontaneous physical activity improves the barrier function and restores SCFA production” creates confusion because the barrier function as such has not been really measured.

We agree with the reviewer comment and have modified in the revised version as follow: Spontaneous physical activity increases tight junction protein expression and restores SCFA production” (line 277).

Crohn's disease mainly affects the ileum. Why microbiota and intestinal permeability determinations have been performed on the colon?

We thank the reviewer for his comment. However, in the animal model used in this study western-blot analysis of CEACAMs expression in CEABAC10 transgenic mice indicated that CEACAM5 was highly expressed in both small intestine and colon, but that expression of CEACAM6 was restricted to the colonic mucosa (Carvalho et al., J. Exp. Med 2009). This confirmed the previously reported lack of ceacam6 mRNA in the ileal mucosa of CEABAC10 transgenic mice (Chan and Stanners, Mol Ther, 2004). In this model, we previously demonstrated that AIEC bacteria are able to colonize colonic mucosa of mice expressing human CEACAMs molecules, and that colonic AIEC colonization may disrupt intestinal barrier integrity before the onset of inflammation (Denizot et al., IBD, 2013). This is the reason why mucosa-associated microbiota and tight junction proteins (occludin and ZO-1) were analyzed on the colonic mucosa.

Reviewer 2 Report

The article’s aim was to assess the preventive effect of physical activity (PA) on the cross-talk between visceral adipose tissue and intestinal microbiota in CEABAC10 mice fed a high fat/high sugar (HF/HS) diet.

Mice were thus divided into wheel group or not, and results showed a decrease in adipose tissue weight after AIEC exposure in the former group. Tight junctions were increased after PA, as well as fecal propionate and butyrate concentration, while microbiota analysis showed an increase in Oscillospira and Ruminococcus in the same group. Authors concluded that spontaneous PA might promote healthy gut microbiota composition changes in mice models that mimic Crohn’s disease, fed a HF/HS diet.

Although the design of the study might be potentially interesting, data reported do not represent a substantial advance in the field. Moreover, the study, in the present form, shows some major flaws.

·       One point of weakness of this article is the inability to correlate the results with the intestinal inflammatory state. The lack of comparison between the inflammatory state before and after AIEC administration is a fundamental issue that cannot be ignored. It should be stressed whether inflammatory changes at the end of the experiment occurred due to AIEC infection or to the different PA, since this is a fundamental step to draw all the conclusions;

·       Histology and cytokines level in the lamina propria of the intestine should be analyzed and reported, for the same purpose.

·       By studying the level of inflammation before and after AIEC administration, the authors could have added an important data, namely the ability of PA to influence the susceptibility to AIEC infection.

·       Fecal microbiota composition was analyzed post-mortem and conclusions were postulated based on the differences between the two groups only at this time point. We do not know, from the present paper, whether the same differences existed before AIEC administration, and this is crucial to say that PA and AIEC are linked to the post-mortem differences found.

·       Page 5, line 233: the criteria used to divide the wheel group mice into “almost inactive, moderately active and active” have not been sufficiently defined.

·       Page 6, line 238: speed was positively correlated with total lean mass, but the effects on total fat mass have not been mentioned.

·       Page 7, line 264-266: “body weight loss (percentage) was not significantly different between the two groups, although it was lower in mice that run faster”, this deserves to be discussed in the discussion section.

·       Page 7, line 266-267: The assumption that visceral fat mass is equal to mesenteric fat is not totally correct, most of the visceral fat in male mice comes from epididymal fat. Moreover, the inflammatory state of visceral fat has not been studied and reported in this study, thus we cannot assume that it is the same between the two kinds of fat.

Minor concerns:

·       Page 8, line 276: it is written “HF/HS diet alter” instead of “HF/HS diet alters”;

·       Page 10, line 333: it is written “and its mainly due” instead of “and it is mainly due”

Author Response

Point by point replies to the Editor and to the Reviewers’ comments

We are grateful to the Chief Editor and the Reviewers for their thoughtful and constructive comments. Below, we have replied to the issues raised in a point-by point manner, and we revised our manuscript accordingly. Thank you for considering this revised manuscript.

Comments and Suggestions for Authors: reviewer 2

The article’s aim was to assess the preventive effect of physical activity (PA) on the cross-talk between visceral adipose tissue and intestinal microbiota in CEABAC10 mice fed a high fat/high sugar (HF/HS) diet.

Mice were thus divided into wheel group or not, and results showed a decrease in adipose tissue weight after AIEC exposure in the former group. Tight junctions were increased after PA, as well as fecal propionate and butyrate concentration, while microbiota analysis showed an increase in Oscillospira and Ruminococcus in the same group. Authors concluded that spontaneous PA might promote healthy gut microbiota composition changes in mice models that mimic Crohn’s disease, fed a HF/HS diet.

Although the design of the study might be potentially interesting, data reported do not represent a substantial advance in the field. Moreover, the study, in the present form, shows some major flaws.

1/ One point of weakness of this article is the inability to correlate the results with the intestinal inflammatory state. The lack of comparison between the inflammatory state before and after AIEC administration is a fundamental issue that cannot be ignored. It should be stressed whether inflammatory changes at the end of the experiment occurred due to AIEC infection or to the different PA, since this is a fundamental step to draw all the conclusions;

We understand the comment of the reviewer. However, in this study, we have chosen not to use either DSS or antibiotics that allow AIEC to colonize better and induce more severe intestinal inflammation. In this model, exposure to AIEC combined with the HFD regimen was used to reflect what is happening in patients in remission to study primary prevention of physical activity and its impact on the mucosal-associated microbiota. Besides, we have previously reported using this model that Western diet induces changes in gut microbiota composition, alters host homeostasis and promotes AIEC gut colonisation in genetically susceptible mice without inducing acute inflammation and histological damages (Martinez-Medina M et al., GUT, 2015). The choice of the model was discussed lines 321-328.

Furthermore, we monitored intestinal inflammation before AIEC infection by assaying two markers of inflammation, fecal lipocalin 2 and chitinase 3-like 1. On these graphs, the HF diet increased, as expected and previously published (Agus et al., Scientific Report,  2016), these two markers (especially chitinase 3-like-1) in the feces between weeks 0 and 12, but without significant differences between the two groups meaning that physical activity did not induce inflammation in this model.

2/ Histology and cytokines level in the lamina propria of the intestine should be analyzed and reported, for the same purpose.

Quantification of some pro-inflammatory cytokines (IL-6 and KC) were performed on colon sections by ELISA, and histological analysis (depth of crypts measurement) was performed after HES staining (see figures below). Since the chosen model did not induce inflammation, we did not observe any difference in these markers between the physical activity and the control groups.

3/ By studying the level of inflammation before and after AIEC administration, the authors could have added an important data, namely the ability of PA to influence the susceptibility to AIEC infection.

As already said, we monitored intestinal inflammation before AIEC infection by assaying two markers of inflammation, fecal lipocalin 2 and chitinase 3-like 1 (fig. point 1). As expected and previously published (Agus et al., Scientific Report, 2016), after the HF diet these two markers increased in the feces between weeks 0 and 12, but without significant differences between the two groups meaning that physical activity did not induce inflammation in this model.

The infection also increased these two markers in both groups. Fecal lipocalin and chitinase 3-like 1 increased between week 12 (before bacterial exposure) and day 2 after AIEC bacterial exposure (see figures below). These data suggest that the AIEC strain could colonize mice gut in the same way between the two groups and that PA does not influence the susceptibility to AIEC infection in this model. In our model (CEABAC10 under high fat diet), the major effect of physical activity was observed on the loss of mesenteric adipose tissue and on the modification of the mucosa-associated microbiota.

4/ Fecal microbiota composition was analyzed post-mortem and conclusions were postulated based on the differences between the two groups only at this time point. We do not know, from the present paper, whether the same differences existed before AIEC administration, and this is crucial to say that PA and AIEC are linked to the post-mortem differences found.

In this study, we focused on the variations of microbiota associated with the intestinal mucosa, we did not analyze the fecal microbiota composition. Therefore, we do not have mucosal specimens for mice before exposure to AIEC bacteria. What interested us in this model was to compare a spontaneous PA model using wheel activity versus controls.

5/ Page 5, line 233: the criteria used to divide the wheel group mice into “almost inactive, moderately active and active” have not been sufficiently defined.

We corrected the sentence since the division in three groups was only subjective. We modified by “Analysis of the distance and speed at W12 showed a significant correlation between these two parameters (Figure 2D; r= 0.4; p<0.05)” (lines 232-233).

 6/ Page 6, line 238: speed was positively correlated with total lean mass, but the effects on total fat mass have not been mentioned.

In the present study, speed was not significantly correlated with total fat mass as shown below. This result has been added to the result section (line 238).

Parameter

Speed m.min
  vs. FAT g

Speed m.min
  vs. TA MES

Pearson r

r

-0.2621

-0.1968

95% confidence interval

-0.6233 to 0,1912

-0.5793 to 0.2567

R squared

0.06872

0.03874

P value

P (two-tailed)

0.2510

0.3924

P value summary

ns

ns

Significant (alpha = 0.05)

No

No

Number of XY Pairs

21

21

7/ Page 7, line 264-266: “body weight loss (percentage) was not significantly different between the two groups, although it was lower in mice that run faster”, this deserves to be discussed in the discussion section.

We added this precision in the discussion section (lines 328-330).

8/ Page 7, line 266-267: The assumption that visceral fat mass is equal to mesenteric fat is not totally correct, most of the visceral fat in male mice comes from epididymal fat. Moreover, the inflammatory state of visceral fat has not been studied and reported in this study, thus we cannot assume that it is the same between the two kinds of fat.

We agree with the reviewer comment and we modified in the revised version « visceral » by « mesenteric » fat mass as expected.

Minor concerns:

·       Page 8, line 276: it is written “HF/HS diet alter” instead of “HF/HS diet alters”;

Done, we have modified this mistake.

·       Page 10, line 333: it is written “and its mainly due” instead of “and it is mainly due”

Done, we have modified this mistake.

Reviewer 3 Report

In the present study, Dr Maillard et al examined the preventive effect of spontaneous physical activity on the gut-adipose issue in a mouse model that mimics Crohn's disease susceptibility. They suggested that spontaneous physical activity promotes a healthy composition of gut microbiota and increase short-chain fatty acids production in an experimental mouse model that mimics Crohn's disease, fed with a Western diet. This study is a very interesting and well conducted experimental work that presents useful information on a crucial issue for IBD pathophysiology, as the metabolism of mesenteric adipose tissue.

English language and style are a minor spell check required.

Author Response

Point by point replies to the Editor and to the Reviewers’ comments

We are grateful to the Chief Editor and the Reviewers for their thoughtful and constructive comments. Below, we have replied to the issues raised in a point-by point manner, and we revised our manuscript accordingly. Thank you for considering this revised manuscript.

Comments and Suggestions for Authors: reviewer 3

In the present study, Dr Maillard et al examined the preventive effect of spontaneous physical activity on the gut-adipose issue in a mouse model that mimics Crohn's disease susceptibility. They suggested that spontaneous physical activity promotes a healthy composition of gut microbiota and increase short-chain fatty acids production in an experimental mouse model that mimics Crohn's disease, fed with a Western diet. This study is a very interesting and well conducted experimental work that presents useful information on a crucial issue for IBD pathophysiology, as the metabolism of mesenteric adipose tissue.

English language and style are a minor spell check required.

We thank the reviewer for his comment. We have corrected minor English mistakes by a native speaker.

Round 2

Reviewer 2 Report

No further comments.